# KOSMOS: An Open Source Underwater Video Lander for Monitoring Coastal Fishes and Habitats

**DOI:** 10.3390/s21227724

**Published:** 2021-11-20

**Authors:** Dominique Pelletier, Justin Rouxel, Olivier Fauvarque, David Hanon, Jean-Paul Gestalin, Morgann Lebot, Paul Dreano, Enora Furet, Morgan Tardivel, Yvan Le Bras, Coline Royaux, Guillaume Leguen

**Affiliations:** 1Ifremer, Unité Ecologie et Modèles Pour l’Halieutique, Centre Atlantique, F-44311 Nantes, France; 2Ifremer, Laboratoire Détection Capteurs et Mesures, Centre Bretagne, F-29280 Plouzané, France; justin.rouxel@ifremer.fr (J.R.); olivier.fauvarque@ifremer.fr (O.F.); morgan.tardivel@ifremer.fr (M.T.); 3Konk Ar Lab, F-29900 Concarneau, France; david.hanon@gmail.com (D.H.); jean-paul.gestalin@wanadoo.fr (J.-P.G.); paul.dreano@orange.fr (P.D.); furet.enora@gmail.com (E.F.); ghe.leguen@gmail.com (G.L.); 4Fonds Explore, F-29900 Concarneau, France; morgann.lebot@gmail.com; 5Pôle National de Données de Biodiversité, UMS 2006 PatriNat, Station Marine de Concarneau, Muséum National d’Histoire Naturelle, F-29900 Concarneau, France; yvan.le-bras@mnhn.fr (Y.L.B.); coline.royaux@mnhn.fr (C.R.); 6Guillaume Leguen, F-29900 Concarneau, France

**Keywords:** underwater video, monitoring, fishes, benthic habitat, coastal ecosystems, STAVIRO, collective intelligence, open source, citizen science

## Abstract

Background: Monitoring the ecological status of coastal ecosystems is essential to track the consequences of anthropogenic pressures and assess conservation actions. Monitoring requires periodic measurements collected in situ, replicated over large areas and able to capture their spatial distribution over time. This means developing tools and protocols that are cost-effective and provide consistent and high-quality data, which is a major challenge. A new tool and protocol with these capabilities for non-extractively assessing the status of fishes and benthic habitats is presented here: the KOSMOS 3.0 underwater video system. Methods: The KOSMOS 3.0 was conceived based on the pre-existing and successful STAVIRO lander, and developed within a digital fabrication laboratory where collective intelligence was contributed mostly voluntarily within a managed project. Our suite of mechanical, electrical, and software engineering skills were combined with ecological knowledge and field work experience. Results: Pool and aquarium tests of the KOSMOS 3.0 satisfied all the required technical specifications and operational testing. The prototype demonstrated high optical performance and high consistency with image data from the STAVIRO. The project’s outcomes are shared under a Creative Commons Attribution CC-BY-SA license. The low cost of a KOSMOS unit (~1400 €) makes multiple units affordable to modest research or monitoring budgets.

## 1. Introduction

Coastal ecosystems host the richest marine areas in terms of biodiversity, enabling a sustainable livelihood for many human populations around the planet. Yet, they are also prone to both diverse and intense anthropogenic pressures such as fishing, recreational uses, marine renewable energies, pollution, and urbanization in general. Monitoring the ecological status of coastal ecosystems is indispensable to track anthropogenic impacts as well as the outcomes of remediation policies. For fishes, monitoring has been mainly undertaken through either fishing (scientific or commercial) or through diver-operated underwater visual measurements, but underwater optical imagery is being increasingly used as a non-obtrusive and non-extractive observation means for conspicuous biodiversity components [1]. Video-based protocols for fish include Baited Remote Underwater Video (BRUV) landers [2] and Diver-Operated Video (DOV) transects [3]. A potential drawback shared by these techniques is influencing the behavior of the fish either by baiting or the diver’s presence. These potential drawbacks were avoided by sampling with a remote panoramic and unbaited video lander, the STAVIRO [4]. This tool successfully observed fishes and benthic habitats in a cost-efficient manner and with a minimal disturbance and was able to collect replicated observations using a pair of units. By benthic habitat, we mean here abiotic and biotic cover and related parameters [5]. Replicated data are particularly needed in coastal ecosystems where habitat distribution is heterogeneous at a small scale. In addition, because recording occurs over a 360° field of view, fishes and habitats may be quantified over an estimated surface area around the lander. The STAVIRO protocol has proved efficient and reliable with over 5000 valid deployments in varied habitats of both temperate and tropical coral reef ecosystems since 2007 [6]. These contributed to numerous assessments of biodiversity and fished resources in the Pacific, Indian, and Atlantic oceans and in the Mediterranean Sea. In recent years, several autonomous unbaited underwater video systems were developed for specific purposes, e.g., mangroves [7], trap cameras for deep species [8], or dark environments [9], in general for observations at close distance. 

Despite the STAVIRO’s success, it has become difficult to disseminate and maintain due to the turnover of camera models, the dependence on parts distributed by specific brands or makers, and discontinued products. This was problematic for ensuring the consistency of observations over time and across lander generations, e.g., for long-term monitoring needs. For the same reasons, it was difficult to fabricate additional landers required for broad-scale monitoring where standardized data are collected at regional scale by several operators. More generally, there is a vivid willingness to share data and standard operating procedures, and facilitating the adoption of the protocol by a larger number of users calls for a new lander design based on generic components that are reliable, easily sourced at a reasonable cost, and that will be available for many years.

In this paper, we present a video system that meets these design concepts: the KOSMOS lander. The three main aims for the prototype were low cost, reproducibility, and full Open Source documentation, i.e., with all technical and engineering detail to be available as Open Source data. Unlike the STAVIRO, the KOSMOS incorporates independent parts including a lens, sensor, electronics, housing, and connectors to replace the camera. The design and fabrication of the KOSMOS are entirely new concepts, based on the previous STAVIRO system and the experience of 12 years of successful implementation, but incorporating new enabling technology and the collective intelligence of a suite of technical experts.

This work was achieved outside of academia or industry within a FabLab, i.e., a digital fabrication laboratory providing access to the environment, skills, and materials for people to learn from other’s expertise and stimulate innovation (https://fabfoundation.org/getting-started/#fablabs (accessed on 14 October 2021)). A FabLab relies on volunteers and aims at outreach, and as such implements collective intelligence and fosters citizen science. 

In this paper, we firstly described the required specifications and operational needs for the lander based on the STAVIRO experience. We then describe the approach and the prototype (hardware and software). Results from tests in a pool and an aquarium are reported, and lastly, we discuss this project from the standpoints of both the technological outcomes and the collective intelligence process.

## 2. Materials

### 2.1. Specifications of the STAVIRO System

The parent system consists of two watertight housings connected by a stainless-steel axis (Figure 1). The upper housing is a plexiglass cylinder with a flat window at one end, and at the other end, an aluminium lid secured by stainless steel crews and bolts. It contains the camera and the electronics, allowing the camera to be switched on and off with a magnet to activate a reed switch through the housing. The specifications for the camera are a high definition (Full HD, i.e., 1920 × 1080 pixels), an approximate field of view of 60°, a large sensitive low-noise back-illuminated sensor (SONY™ CMOS Exmor R sensor), and a capture rate of at least 25 frames per second in a progressive scanning system (25p). Higher definitions or capture rates are possible but inflate file size. The camera focus is manually set to infinity to ensure the observation of fishes at a distance of at least 5 m. Fishes are typically observed from very close to the housing until distances that may extend to more than 10 m when underwater visibility is excellent, which regularly happens in coral reef areas.

The lower housing encloses a motor with electronics and a battery. The motor is related to the stainless-steel axis that enables the upper housing to rotate at programmed angles and timings. Following extensive testing in 2007 and 2008, the program was set to rotate the camera housing 60° every 30 s, so that six contiguous 60° fixed frames of 30 s are recorded for each 360° rotation; the duration of a rotation is hence ~3 min. 

The device is fixed on a three-leg aluminium stand used for deployments. The stand is equipped with an intermediate buoy that tenses the rigging to avoid mingling with the upper housing. That buoy is itself fixed to a line with a large float at the surface to spot the lander and retrieve it when necessary. Each leg is weighted with 2 kg of lead, and a depthmeter is fixed on one of the legs. 

The STAVIRO lander is dropped and retrieved from a small boat and has been used in depths ranging from 1 to 80 m. It does not have artificial light to avoid influencing animal behavior. It has been used in many different weather, wave, and current conditions. The lander is set horizontally on the seabed at the desired location. One observation consists of three complete undisturbed rotations (~9 min). To minimize disturbances due to boat presence, engine noise, and lander drop and retrieval, the lander is left in situ for approximately fifteen minutes. Deployments are conducted using two devices alternatively to optimize the time spent at sea. 

The STAVIRO is a lightweight lander that can be carried and manipulated by one person. After disassembling, it is usually transported in a reasonably small protective case such as a Peli^TM^ case. It was thus designed for an easy use, enabling the collection of numerous observations per day at sea.

An automated version of the STAVIRO was also developed, the MICADO, which is deployed for several days to record at programmed intervals.

The STAVIRO and MICADO landers have captured thousands of full HD footages of fishes and their habitats using the AVCHD™ format, which is based on the MPEG-4 AVC/H.264 for image compression.

### 2.2. Specifications for the KOSMOS System

#### 2.2.1. Optics

An essential feature for the new system is that the images are fully consistent with the images previously collected by the STAVIRO. First, the optical system must record high definition (HD) images, i.e., 1920 × 1080 pixels, to have the same precision to identify and count fishes. Second, image capture must follow the principle of six successive frames of 30 s with a horizontal field of view (FOV) of 60°. Due to the seawater refraction index correction, this field of view in water corresponds approximately to 84° in the air. Third, a good depth of field (DOF) is required from 1 to 5 m at least because the STAVIRO is aimed to observe fish in the distance, visibility permitting. Fourth, image distortion must be minimized to facilitate species identification and distance and size estimation. A critical fifth parameter is the objective’s numerical aperture, which allows adjusting the trade-off between the sharpness of the image and the exposure time. Ambient light at 10–40 m depth is very low. Other low-light conditions include image capture at sunrise or sunset, which are of particular interest for fishes and marine fauna. To maximize light, the objective’s aperture must be large, and a sensitive sensor is needed. When depth increases, images may be post-processed to improve color rendering. 

#### 2.2.2. Mechanical Integration

Mechanical integration is mostly constrained by the dimensions of the components to be enclosed. The system must also remain compact to minimize underwater volume and to be easily handled during field work and transportation. From our experience, the housing should be equipped with a valve for easing its opening in the case of temperature gradients and for testing watertightness.

#### 2.2.3. Power, Electronics, Sensors, and Software

Rotating the camera and recording and saving images both require energy. Currently, the STAVIRO autonomy is approximately six to seven hours for the camera and ~15 h for the motor. Since field work lasts approximately six hours per day, this autonomy means that there is no time wasted nor water-related risk inherent to changing batteries on the boat during the day. While the STAVIRO autonomy is enough for routine use, its automated version would benefit from a larger autonomy.

All electronics boards and components must have a reduced size in order to keep the KOSMOS small and compact. Compactness is a central criterion for selecting the microcomputer, pressure and temperature sensors, camera and its optics, a real time clock module, and LED indicators. The data flow must be controlled with high-speed writing between the operating system installed on the microcomputer and the storage device. 

The KOSMOS requires temperature and pressure sensors for image processing (color correction) and as auxiliary information for data analysis. Lastly, the date and hour of shot are essential to manage the recorded images. These measurements must all be saved in a log file. A GPS was not deemed indispensable as the location of the lander may be measured independently.

## 3. Methods

### 3.1. Infrastructures and Tools Used for Prototype Conception and Reproduction

Most of the technical work was conducted at the Konk Ar Lab (KAL) FabLab, which has workshops, tools and welding equipment, electronics, and mechanics. The digital tools used for prototyping some of the parts include 3D printers and the Fusion 360° software (Autodesk Inc., San Rafael, CA, USA, https://www.autodesk.com/products/fusion-360/personal (accessed on 14 October 2021)) for design and modeling. Mechanical parts and supporting parts were modeled using the Fusion 360°; some were 3D-printed, while others were lasercut, with the advantage of being quick and low cost in both cases. Printed parts were printed in polyethylene terephthalate (PET) with a I3 MK3S 3D printer (Prusa Research, Prague, Czech Republic, https://www.prusa3d.fr/original-prusa-i3-mk3-fr/ (accessed on 14 October 2021)). PET has a good water resistance and solidity. Lasercut parts were cut with poly(methylmethacrylate) (PMMA) or polyoxymethylene (POM) using a RobotSeed RS-1610L Laser Cutter with a 150W laser tube (https://www.robotseed.com/; https://docplayer.fr/179185190-Manuel-laser-decoupeuses-graveuses-laser-robotseed-rs-6040l-rs-1060l-rs-1610l.html (accessed on 14 October 2021)). Programs for recording and image processing were coded in Python [10]. 

Information was shared among the group of scientists and volunteers using a wiki. The documentation and tutorial for KOSMOS reproduction was written and edited throughout the process.

Scientific facilities were used for the optical tests in the laboratory, for the tests in a controlled pool (Section 3.3 and Section 4.7), and for the tests in the aquarium (Section 4.8). The latter was conducted in the Concarneau Marinarium, which has large transparent tanks with fishes from Atlantic species in their natural habitat (https://www.stationmarinedeconcarneau.fr/en/visit-marinarium/spaces-marinarium-2356, accessed on 14 October 2021). The ImageJ software [5] was used for analyzing the images from the optical tests.

### 3.2. An Approach Based on Collective Intelligence

The conception and development of the KOSMOS proceeded over a 16-month period of time from May 2020, with ten regularly planned workshops and between the workshop working sessions in subgroups. All the workshops took place at Concarneau’s Fab Lab, Konk Ar Lab. An initial workshop aimed at describing the project and requirements for the prototype and at identifying and involving volunteers in the team. Subsequent workshops dealt with conception for mechanics, coding, and electronics. Participants with specific skills worked together on the different parts of the system, and meetings were organized to share progress among the participants. A hackathon was organized over a week-end to boost the construction of a first prototype.

The version 1.0 of the KOSMOS was trialled at sea in September 2020 with a provisional housing previously used for the MICADO. These tests provided feedback for further development, and in particular for selecting the parts, e.g., the housing. From November 2020, the project had to continue at distance through video calls and remote work due to the country’s lockdown. A virtual network connection enabled volunteers to connect to the Raspberry and continue coding, and some volunteers borrowed the parts and tools to continue the development.

A specific workshop was held to discuss the Open Source license and how to best acknowledge and reward the contributions of each participant in the project. Publishing the present paper was key to this acknowledgement for scientists and for volunteers. In January 2021, a workshop was dedicated to reproduce the prototype from the documentation written by the different people involved. This provided feedback to complete and improve the documentation through better wording and with illustrations. At that same time, the first tests of the KOSMOS in seawater were conducted at the Concarneau Marinarium. From these tests, the first fish images were captured, and several problems were identified in relation to the motor calibration and the watertightness around the switch and sensors on the housing cap. In March 2021, a workshop dealt with the conception and realization of the new rigging system. The endurance of the KOSMOS was then tested by programming a continuous rotation for more than an hour in a freshwater tank. In July and August 2021, the mechanical and supporting parts were completely changed, leading to KOSMOS 3.0; the new parts were then remodeled, and some of these parts were produced using laser cutting machines. Tests in a pool were conducted in April and September 2021 at the Ifremer facility in Lorient to assess the optical performance of the KOSMOS camera compared to the cameras used for STAVIRO.

The work of volunteers at the FabLab was coordinated by an independent worker who was also involved in some of the developments, while the overall coordination of the project was shared with the lead scientist. The technical coordination of the project represented 1350 h of work. 

### 3.3. Experimental Set-Up for the Tests in a Controlled Pool

Experimentations took place at Ifremer in a freshwater pool 12 m long, 2.6 m wide, and 1.5 m deep. The flume pool is a hydrodynamic facility available for any qualification and observation of submerged objects in still or moving water; during these tests, the water was still. 

The test pattern had several black and white grid patterns of different sizes to calculate the cameras’ FOV and image distortion, and a color chart to evaluate color rendering (Figure 2a).

Three systems were tested: the KOSMOS prototype, a STAVIRO equipped with a SONY™ CX900E camera (STAVIRO-Sony), and a STAVIRO equipped with a Paralenz Dive Camera+ (STAVIRO-Paralenz, Paralenz, Rødovre, Danemark, https://www.paralenz.com/ (accessed on 14 October 2021)). Both STAVIRO configurations have been used in recent years.

Each of the three systems was successively placed at the following distances: 1, 2, 3, 4, 5, and 6 m from the test pattern (Figure 2b). Distances were measured with a decameter. Each camera recorded images from the patterns and images were extracted from the video in order to be analyzed.

#### 3.3.1. Field of View Estimation 

The FOV θ (in degrees) was calculated from Equation (1):(1)$$\theta =2\;arctan\left(\frac{N_{pixel\;x}\;dx_{cm}}{2\;D\;dx_{pix}}\right)*\frac{180}{\pi },$$
where $dx_{cm}$ is the horizontal distance (in cm) between the two edges of the pattern test, $dx_{pix}$ the same distance (in number of pixels), measured with ImageJ software, $N_{pixel\;x}$ the horizontal size of complete image (in number of pixels), and D the distance (in cm) between the camera and the pattern. 

#### 3.3.2. Image Distortion and Correction

The optical distortion is both due to the non-ideal nature of the objective and to the use of a flat port. Imaging deformations associated with this optical system are well-described [11]. In this experiment, we compared the overall distortion of the three systems. To calculate distortion, we used a function that transforms a regular grid to a distorted one depending on a distortion coefficient *d*. This function brings nearer or further the grid nodes depending on *d* and their distance to the image center and is described by Equations (2) and (3):(2)$$x_{\mathrm{distorted}}=x_{\mathrm{real}}+d*{\left(x_{\mathrm{real}}{}^{2}+y_{\mathrm{real}}{}^{2}\right)}^{1/2},$$
(3)$$y_{\mathrm{distorted}}=y_{\mathrm{real}}+d*{\left(x_{\mathrm{real}}{}^{2}+y_{\mathrm{real}}{}^{2}\right)}^{1/2},$$
where $x_{\mathrm{real}}$ and $y_{\mathrm{real}}$ are the spatial coordinates of the regular grid, while $x_{\mathrm{distorted}}$ and $y_{\mathrm{distorted}}$ correspond to the coordinates after distortion. Figure 3 illustrates a barrel (b) and a pincushion (a) distortion.

The distortion coefficient *d* is estimated by fitting the distortion function to the *x* and *y* coordinates of the image of the test pattern nodes obtained for each camera. *d* documents the nature (barrel or pincushion) and quantifies the magnitude of the distortion. *d* is also needed to post-process images to correct for distortion by using the inverse distortion function.

#### 3.3.3. Color Rendering

Colors are composed by RGB coefficients (R, G, and B), which correspond to the proportion of red, green, and blue, respectively on a pixel image. Because of water light absorption, the RGB coefficients decrease as a function of capture distance. This attenuation also depends on the properties of the sensors and lenses and was measured as follows. From the videos recorded by each camera, images were extracted with the VideoLanC (VLC) software [12] for each distance from 1 to 6 m (by 1 m step). RGB coefficients were measured using the pipette function of the Microsoft PowerPoint© software. Color rendering was evaluated by comparing the coefficients for each camera to those from the standard color pattern.

## 4. Results

The KOSMOS prototype (Figure 4) was assembled from the components listed in Table 1.

### 4.1. Optics

The optical system consists in a 12 Megapixel (Mp) Raspberry Pi camera (PiCam) HQ 12MP (Table 1), a 4 mm focal length AICO objective with a F-number of F/2.0 adapted for 1/2″ PiCam sensor. The sensor diagonal is 7.9 mm, and the lens with the objective has a FOV of 82° in air. Once corrected for seawater refraction, the FOV in water is 59°. The AICO lens is fixed on the PiCam using a standard C-mount ring (Figure 5).

### 4.2. Housing and Integration of Components

From our experience with the STAVIRO, the cylinder shape is the most appropriate, as it enables equal pressures on the housing surface, and in addition, a clear housing is necessary for checking the functioning from outside. This cylinder configuration provides degrees of freedom to optimize the arrangement of the different parts and supports within the tube.

The KOSMOS housing is a Blue Robotics 298 mm long transparent acrylic tube with a 4 inch diameter. Blue Robotics specializes in open source material for underwater vehicles or robots, which ensures the part is easily available and will be continued. The dimensions of the tube determine those of the optical and electrical components to be enclosed. The tube is closed by an aluminum opaque cap at one end and a 5 mm thick clear acrylic window at the other end. Both ends are secured by four screws and double O-ring flanges for watertightness.

In the final version (KOSMOS 3.0), all system components are assembled within the housing through four supporting parts, which were designed and then modeled using the Fusion 360° software and printed with a 3D printer (Figure 4 right). Because the parts were small, a 100% filling rate was used for 3D printing. These parts are fastened together with screws, and the entire system slides within the housing for easy manipulation but cannot rotate so that the camera is always horizontal within the housing.

### 4.3. Rotations of the Housing

Rotations are by steps of 60° every 30 s. A marinized engine located outside of the housing was preferred, mostly because there was no risk of water leaking in through the rotating axis. In addition, this left more space within the housing for other components. The KOSMOS rotation is operated by a brushless motor F2838-350KV (Blue Robotics Inc., Torrance, CA, USA). The motor speed had to be reduced from 2500 to 14.7 rpm to satisfy the parameters of the rotations. A motor reducer was designed and modeled using the Fusion 360° software. It works with four couples of gears (Figure 6), resulting in an overall reduction ratio of 1/143.8. A Maltese cross (Figure 6) was added to the reducer to ensure each single rotation is exactly 60°. The reducer and Maltese cross are laser-cut in 5 mm PMMA sheets; they may also be cut in POM sheets. After each rotation, a reed switch fixed on the support of the reducer aligns with a magnet fixed on the drive wheel and stops the motor. When the motor stops, it is immediately ready for the next rotation, which is controlled by the code.

The housing is clamped on the reducer using pliers and stainless-steel threaded rods and nuts (Figure 4 right). Pliers are laser cut in a PMMA sheet. The housing is placed between the pliers, centered above the reducer and fastened so that its gravity center does not move during rotation. Power is supplied by the battery to the motor and reed switch through two electric cables (Figure 4 right).

### 4.4. Support and Rigging for Deployment

The KOSMOS is dropped in water and retrieved using a tripod support and a rigging system (Figure 4 left). We reused the anodized aluminum support made for the STAVIRO, with an additional part to keep the same elevation of the housing with respect to the sea floor, since the KOSMOS has one housing instead of two for the STAVIRO. The support is rigged to an intermediate buoy that keeps the rigging tight; this buoy will itself be fixed to a line connected to a large float at the surface that will be used to spot the system and retrieve it when needed. The KOSMOS housing is longer than the upper housing of the STAVIRO. To avoid the entanglement of the housing in the rigging, three 70 cm long ropes in Dyneema^®^ material (Ino-Rope, Concarneau, France) were designed and fastened on the support at one end and fixed to three aluminum arms (30 cm long and 120° angle between two arms) at the other end. The subsurface buoy was directly secured at the intersection of these arms.

### 4.5. Electronics

All electronic components are connected to a Raspberry Pi 4 (RPi4) (Figure 7). This single-board computer was selected because of its low cost, size, modularity, and the ability to support USB and HDMI. The power is supplied by a Lithium Polymer battery LiPo 3S 2200 m Ah. The 12 V voltage delivered is converted into 5 V to run the RPi4. The system design allows to have four batteries in parallel to increase power autonomy. The brushless motor F2838-350 KV is supplied with 12 V power and its driver BF32 23A with 5 V power (mono canal relay). The RPi4 controls the brushless motor with the relay. Raspberry Pi Raspbian OS is set up in the microcomputer and a USB key is used for the data storage.

Several sensors and light indicators are connected to the RPi4: (i) a real time clock (RTC) DS3231 AT24C32 to a saved date and time for each recording; (ii) a pressure and temperature sensor Bar30; (iii) LEDs to indicate the working state of the system; and (iv) reed switches to switch the RPi4 on and off, to record data, and to stop the motor after 60° KOSMOS rotation.

### 4.6. KOSMOS Recording Process

The recording process is in four steps (Figure 8). During the starting step, the system is powered using the watertight switch placed on the end cap. The recording system boots on the Raspbian OS, the RPi4 is initialized, the time is set with the RTC module, and a CSV file that will receive the time, pressure, and temperature measurements is created. The brushless motor is then armed. A blue LED on the secondary electronic board (Figure 7, top left) is flashing during the starting step and becomes still to inform the operator that the initialization stage is over. The next step (working step) starts when the operator engages the recording process by using a magnet to activate the start/stop reed switch. This causes the blue LED to switch off and starts the rotation of the motor to align with the magnet located on the drive wheel. From this moment, the housing rotates 60° every 30 s. Image data are continuously recorded in a H264 video file on the USB key during the working step.

At the end of the desired recording duration, the operator uses a magnet to activate the start/stop reed switch in order to stop the process (stopping step). The recording and the motor are thus set in standby. The blue LED switches on in still mode to indicate that the system is ready for the next recording. To record a new video, the operator will have to use the magnet again to activate the start/stop reed switch.

At the end of the field work, the operator uses the magnet to activate the start/stop reed switch in order to shut down the KOSMOS (shutdown step). All metadata (date, time, pressure, and temperature) are then written on the CSV file and transferred to the USB key before the system’s shutdown.

For the maintenance and parameterization of the KOSMOS program, the RPi4 is mounted on a laptop with a Unix terminal. Secure SHell (SSH) communication is used to make any changes to the code and operating parameters.

### 4.7. Experimental Results

#### 4.7.1. Field of View

FOV were calculated for the KOSMOS, STAVIRO-Sony, and STAVIRO-Paralenz systems from images (Appendix A) from Equation (1); they were, respectively, 59.3°, 63.6°, and 69.4°. With a FOV of 59.3°, the KOSMOS system was very close to the desired specifications. The Paralenz has a larger FOV, probably because this waterproof camera designed for divers uses several lenses.

#### 4.7.2. Image Distortion and Correction

The distortion coefficients were estimated at a 1 m distance (Figure 9, left). The coefficients were, respectively, 0.00013 pix^−1^ for the KOSMOS, 0.00015 pix^−1^ for the STAVIRO-Sony, and −0.00005 pix^−1^ for the STAVIRO-Paralenz. Hence, the KOSMOS and STAVIRO-Sony display similar moderate pincushion effects (Figure 3), while the STAVIRO-Paralenz exhibits a slight barrel distortion (Figure 9, right and Appendix A). The distortion coefficient of the KOSMOS enables the correction of images, showing that the distortion is low and results in a slightly non-squared FOV (Figure 10).

#### 4.7.3. Color Rendering

Color rendering for the KOSMOS 3.0 camera was good (Figure 11) and corresponding RGB coefficients at 1 and 3 m distances were close to those from the results of the STAVIRO-Sony and the STAVIRO-Paralenz (Appendix A). During the early tests conducted in April 2021 (not reported), color rendering was better than in September 2021 because the distance between the camera objective and the window was larger in KOSMOS 2.0 than in KOSMOS 3.0. In the next version, the position of the objective with respect to the window will be optimized to improve color rendering.

For distances ranging between 1 and 3 m, colors with a high wavelength (infrared to orange) were unsurprisingly more absorbed, while green and blue colors were less affected. At a 6 m distance, color rendering was altered for all three cameras despite the absence of water turbidity.

### 4.8. Test with Fishes

The most recent fish images were captured in the seawater aquarium at the Marinarium in October 2021 using KOSMOS 3.0 (Figure 12). The objective of the test was to check the quality of the images recorded by the prototype. Screenshots in an aquarium give a realistic example of the images that will be captured in real conditions, with no artificial light. Together with the videos of fish silhouettes recorded during the pool tests in September, they illustrate the quality of the underwater images captured by the KOSMOS camera in conditions of high visibility (Figure 13).

The real fish are perfectly identifiable at close distance (1–3 m) despite their swift movements, and the fish silhouettes appeared precisely delineated over the 1–6 m distance range, confirming the DOF of the KOSMOS camera.

## 5. Discussion

### 5.1. Meeting the Initial Specifications

The KOSMOS prototype satisfies the requirements, and primarily the optical performance, which makes the videos collected consistent with those previously collected by the STAVIRO. The FOV of 59°, DOF, image quality, color rendering, and a low image distortion fully meet the specifications, except for the image size (this point is discussed below). Reaching this objective through the search and assembly of a lens, a sensor, and a recording device was a genuine challenge at the onset of the project. The collaboration between specialized scientists and volunteers with technological skills and interest in underwater photogrammetry was essential for this to happen. It must be noted that finding the proper objective required an extensive search.

The size of the images captured by the KOSMOS is 1600 × 1200 (UXGA, 1.92 Mp), corresponding to an aspect ratio of 4:3, while for both STAVIRO cameras, image size is 1920 × 1080 (Full HD, 2.07 Mp), i.e., an aspect ratio of 16:9. This resulted from the native size of the PiCam sensor, which is a 4:3 12 Mp resolution recording 1600 × 1200 videos. UXGA is actually the 4:3 image format that is closest to Full HD. The images from the tests show that this difference in aspect ratio should not compromise the ability to identify and count fishes. We expect that, in the near future, it will be possible to find a full HD sensor that will result in a lens–sensor combination with the same FOV as the STAVIRO. 

In other respects, the prototype actually exceeds the expectations in terms of image capture. First, the image format facilitates post-field image correction with respect to depth-dependent color absorption (an interesting feature of the Paralenz camera) and turbidity [13,14,15], among other factors. In a similar way, image distortion was evaluated, and images may be corrected. Developing a code for the correction of images is envisaged as a perspective of the project. Color rendering will even be improved with respect to the STAVIRO-Sony once supporting parts will have been modified to bring the objective closer to the window. Second, temperature and depth are automatically recorded in auxiliary files. Third, with a single housing, the KOSMOS is simpler and more compact than the STAVIRO. Camera parameter settings are permanent, thereby avoiding the need to check them before field work, which avoids human errors and also undesirable changes in settings that sometimes happened with used cameras. This simplicity is an advantage for citizen science and for any deployments in general, as it contributes to the repeatability in system reproduction and use.

The extended experience with the STAVIRO in terms of field implementation and image analysis was another important asset. It gave a very precise scope for devising the prototype and making technological choices, and it enabled focusing on the essential features for the KOSMOS while forgetting about the unnecessary ones. This experience also helped additional improvements aimed at easing field work and increasing autonomy. 

At the end of the prototyping phase, the TRL of KOSMOS 3.0 was 6. Next stages include the reproduction of several units and extensive testing on the field, in order to reach TRL 8. Future field work at sea will include testing the optical performance of the KOSMOS in varying conditions of visibility and light.

Last, the way the KOSMOS was conceived and made and the rapid advancement of low-cost imaging technology (e.g., https://www.stereopi.com/v2 (accessed on 14 October 2021)) allow envisaging other developments such as the programmable version of the STAVIRO (MICADO) [16,17] and a stereo version that will provide size-based information and distance measurements. The MICADO is particularly relevant in places where visibility varies, as it can record images at planned timings.

### 5.2. Reproducibility and Cost

The cost of the lander is approximately 1360 € (rigging and support not included), thus making this system affordable to modest research or monitoring budgets. It is not a low-cost system, but rather a reasonably priced one, in particular with respect to its high-quality optical performance. In this overall cost, the objective and the watertight connectors were expensive in comparison to the other components. The lander components may be bought online from popular companies. We found a single objective able to meet the FOV specification in combination with the PiCam but alternative models are likely to become available in the near future.

Several minor amendments are envisaged or underway to facilitate the reproduction, such as using printed circuit boards rather than making the boards.

The quality of the documentation for reproduction is being tested during workshops where other systems are built by other volunteers. The documentation is available at https://wikifactory.com/@konkarlab/kosmos30 (accessed on 14 October 2021). It is, at the moment, only available in French. Translation into English will be achieved shortly, as the documentation is currently being finalized for dissemination.

Last, it must be noted that the KOSMOS may also be made with higher-grade components (at the expense of a higher cost), e.g., for the purpose of long-term monitoring or research. For instance, an industrial objective and camera may be used, or some parts may be made from stainless steel.

Several systems able to capture underwater video for observing biodiversity were proposed in recent years that share similar low-cost and do-it-yourself objectives (Table 2). KOSMOS mostly differs from other propositions in that it is the successor of an existing system used for a long time.

### 5.3. Open Source and Digital Fabrication Laboratory

The project was mostly conducted within a FabLab. We could find no published example of collaboration between a FabLab and environmental researchers to devise and construct an observation system for collecting high-quality images. The existence of a pre-existing operational parent system greatly facilitated the project’s dynamics since most questions raised by makers could be quickly addressed based on the STAVIRO experience. 

This collaboration has several advantages. First, it facilitates the dynamics of a project, as (i) administrative constraints are minimal and (ii) creativity and innovation lie at the heart of digital fabrication laboratories. Second, the engagement of both volunteers and scientists results in a productive group guided by a common goal. In terms of citizen science, conceiving and making a technological contribution is a very concrete objective that is bound to maintain the motivation of the makers’ group. These advantages come with some additional efforts. Coordinating the work of participants with different backgrounds, agendas, and motivations requires substantial time and interpersonal skills to avoid misunderstandings between participants and dispersion in activities, while maintaining the momentum of the project. The coordination must also anticipate the availability and potential turnover of volunteers and adapt the team accordingly.

Even if the KOSMOS 3.0 may still be improved, the outcome meets the expected scientific requirements and shows that high scientific standards may be achieved through collective intelligence. Several units will be made in the next months to support a citizen science sampling program in the Concarneau area.

By making the prototype and documentation available, we aim to facilitate the use of the STAVIRO protocol and its broad dissemination. We also expect that with a community of contributors, the prototype may be either improved and/or adopted by other users, including for educational purposes.

In the light of conservation challenges in coastal areas (marine renewable developments, fishing and other anthropogenic pressures), key biodiversity facets and fish resources must be monitored and assessed over large areas and with appropriate spatial replication. The ability to reproduce the KOSMOS at a reasonable cost is an opportunity to meet these requirements and collect numerous images that will be consistent with the existing STAVIRO images. 

## Figures and Tables

**Figure 1 sensors-21-07724-f001:**
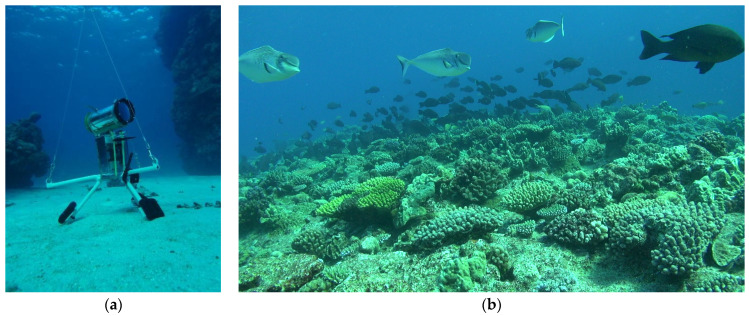
The STAVIRO system deployed to observe fishes and habitats ((**a**), credits: B. Preuss) and example of frame recorded at Astrolabe’s reef, New Caledonia (**b**).

**Figure 2 sensors-21-07724-f002:**
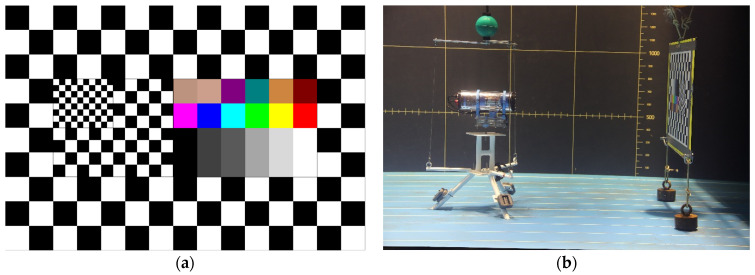
Test pattern use for experimentations (**a**). The KOSMOS system and test pattern in the controlled pool (**b**).

**Figure 3 sensors-21-07724-f003:**
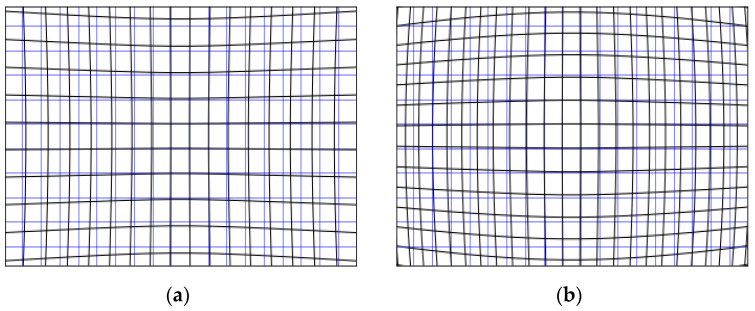
Distorted (in black) and regular (in blue) grids for two distortion coefficients. (**a**): *d* > 0, pincushion distortion. (**b**): *d* < 0, barrel distortion.

**Figure 4 sensors-21-07724-f004:**
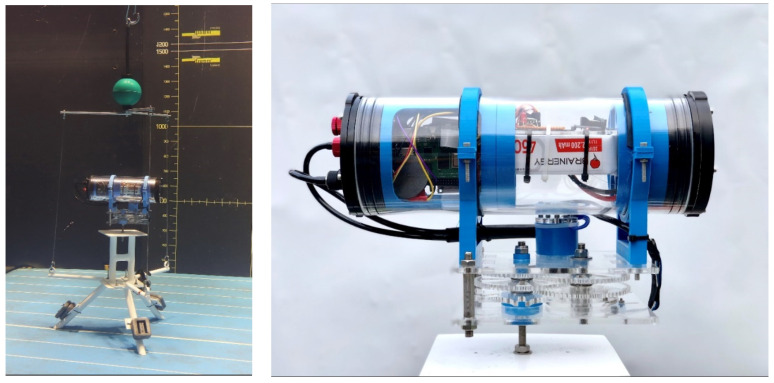
The KOSMOS prototype on its support and with the rigging (**left**). Side view of the housing and reducer (**right**).

**Figure 5 sensors-21-07724-f005:**
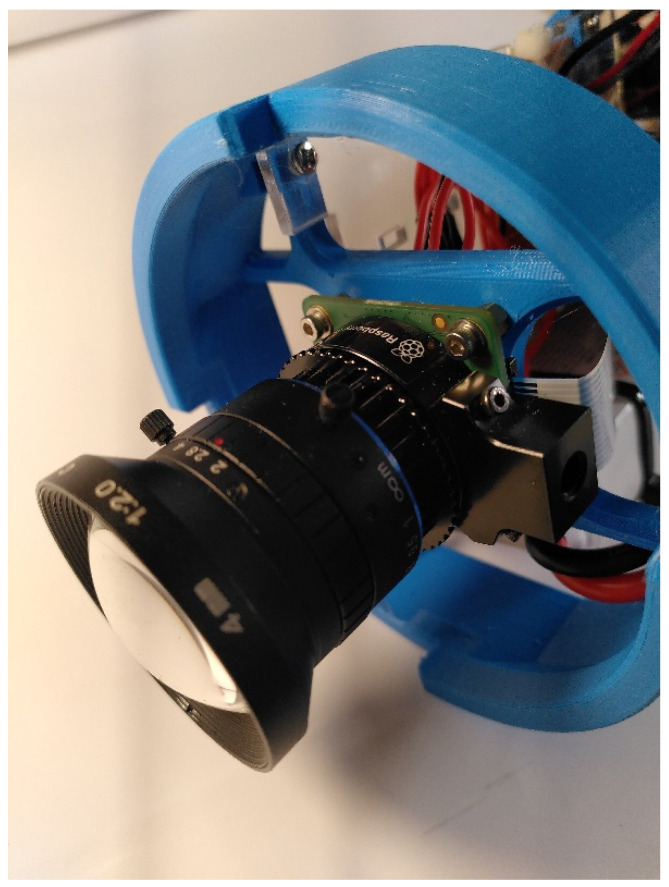
AICO lens with C-mount ring and Pi-Camera HQ 12MP secured on 3D-printed support.

**Figure 6 sensors-21-07724-f006:**
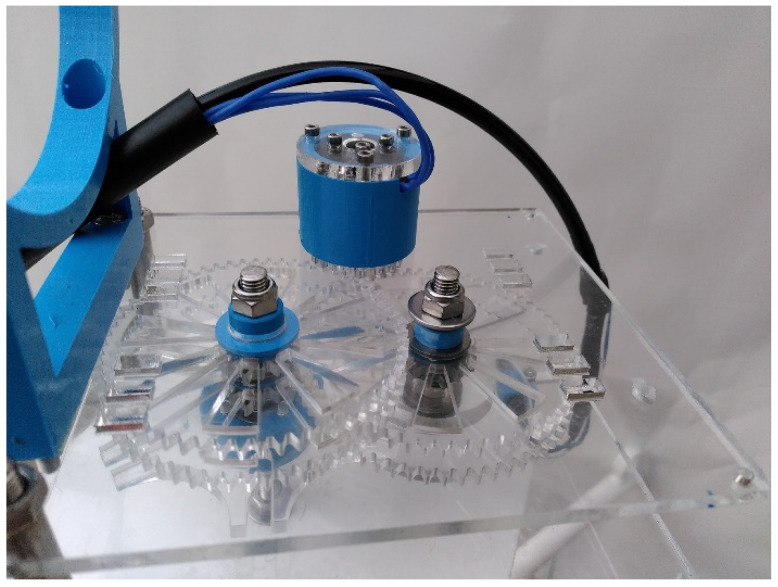
The reducer based on four cogwheels, a drive wheel (**bottom left**), and a Maltese cross (**bottom right**).

**Figure 7 sensors-21-07724-f007:**
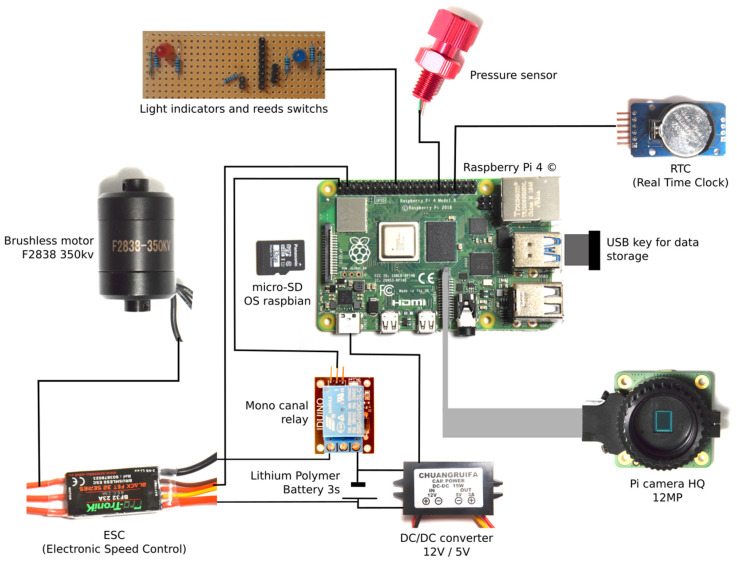
Sketch of the KOSMOS electronics.

**Figure 8 sensors-21-07724-f008:**
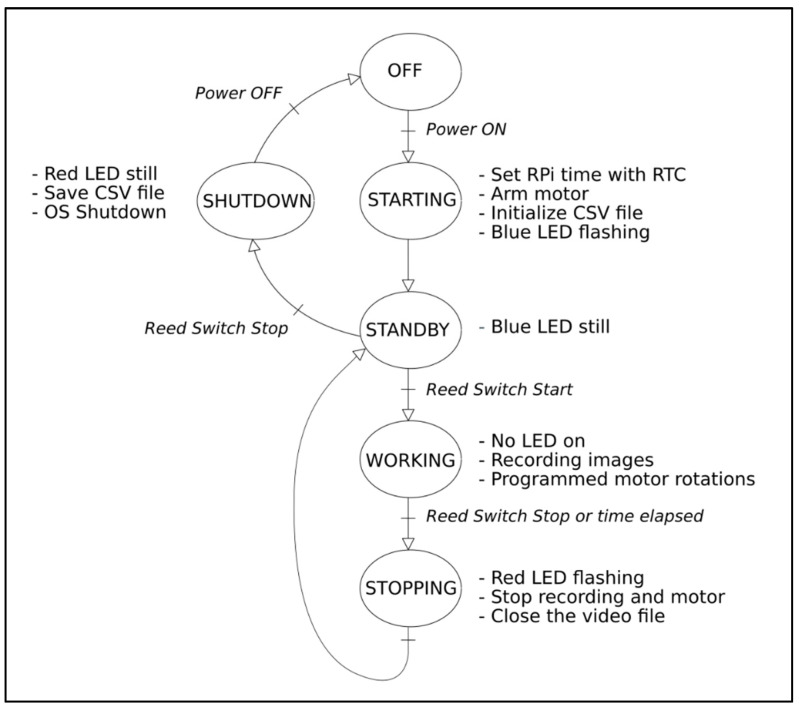
Flowchart of the recording process.

**Figure 9 sensors-21-07724-f009:**
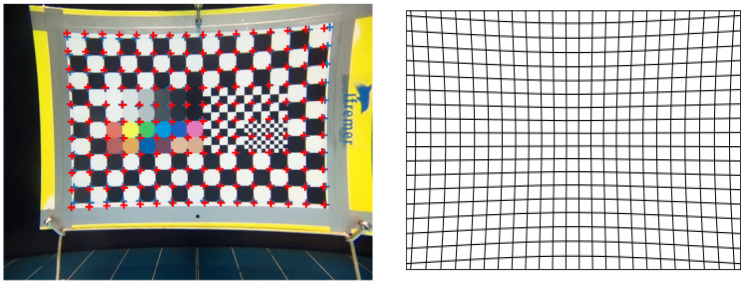
KOSMOS image of the pattern test at 1 m distance. The superimposed blue crosses are the observed nodes of the pattern, while the red are the fitted ones (**left**). Effect of the KOSMOS system on a regular grid (**right**).

**Figure 10 sensors-21-07724-f010:**
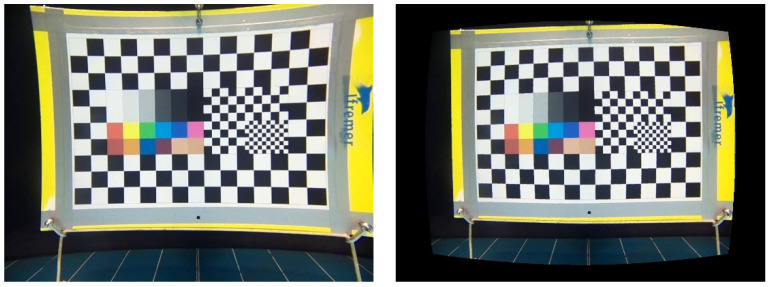
KOSMOS image of the pattern test at 1 m before (**left**) and after (**right**) correcting for distortion.

**Figure 11 sensors-21-07724-f011:**
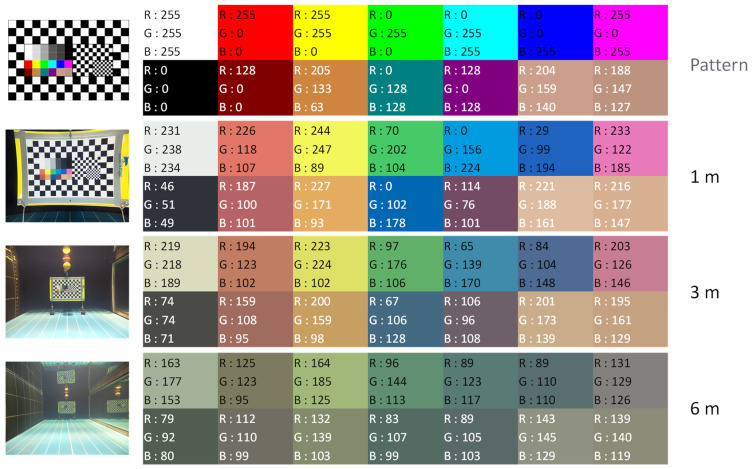
KOSMOS RGB coefficients at distances of 1, 3, and 6 m. Top: RGB coefficients for test pattern.

**Figure 12 sensors-21-07724-f012:**
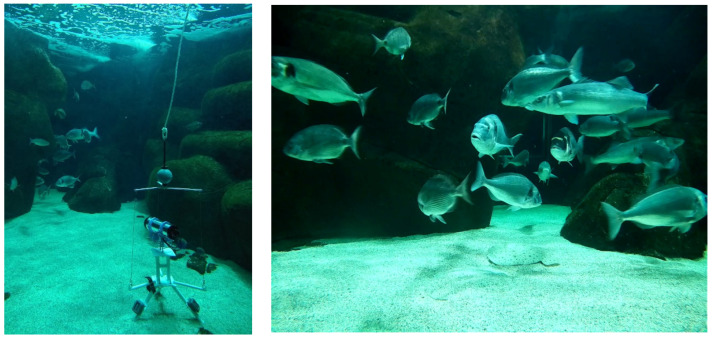
Photo of the KOSMOS 3.0 prototype in the aquarium (**left**). Screenshot of a video recorded in the aquarium in October 2021 (**right**).

**Figure 13 sensors-21-07724-f013:**
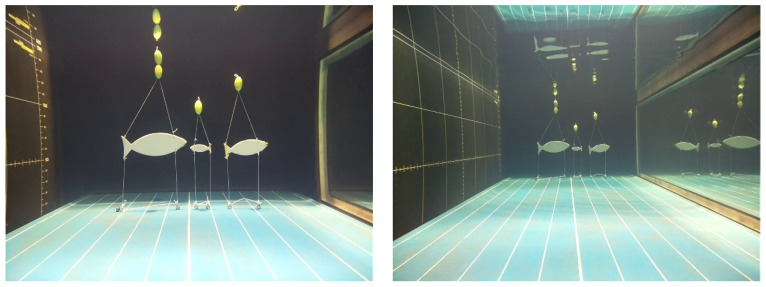
Images of fish silhouettes taken in the Ifremer pool taken with the KOSMOS camera at distances of 3 m (**left**) and 6 m (**right**).

**Table 1 sensors-21-07724-t001:** List of components used for the KOSMOS prototype. The parts for the support and rigging are not detailed here.

Part	Reference	Manufacturer
**Body tube**	WTE4-P-TUBE-12-R1-RP	Blue Robotics Inc., Torrance, CA, USA
**Aluminium end cap, five holes**	WTE4-M-END-CAP-5-HOLE-R1-RP	Blue Robotics Inc., Torrance, CA, USA
**Acrylic end cap**	WTE4-P-END-CAP-R1-RP	Blue Robotics Inc., Torrance, CA, USA
**Camera**	Raspberry Pi camera HQ 12 MP	The Raspberry Foundation, Cambridge, UK
**Objective**	ACH0420MM	AICO Electronics Limited, Hangzhou, China
**Micro-computer**	Raspberry Pi 4	The Raspberry Foundation
**DC/DC converter**	Car power DC/DC 15 w 12 v to 5 v	Chuangruifa, Shenzhen, China
**Brushless motor**	F2838-350 KV	Blue Robotics Inc., Torrance, CA, USA
**Pressure and temperature sensor**	Bar30	Blue Robotics Inc., Torrance, CA, USA
**Electronic speed control**	BF32 23A	Protronik, Saint-Martin-d’Hères, France
**Relay**	SRD-05VDC-SL-C	Shenzhen Jiayuancheng Electronics, Guangdong, China
**Lithium Polymer battery**	Brainergy BY.2200.3S.45-XT60, 2200 mAh	Yuki models, Bad Bramstedt, Germany
**Micro SD card**	32Go writing speed 10 minimum	Integral Memory, London, UK
**USB key**	64 Go	SanDisk, Milpitas, CA, USA
**Two cobalt connectors**	3 pins (12A)	Blue Trail Engineering, Longmont, CO, USA
**RTC module**	DS3231 AT24C32	iTeadStudio, Shenzhen, China

**Table 2 sensors-21-07724-t002:** Comparison of underwater video systems with similar uses. For KOSMOS and STAVIRO, the weight includes the tripod and 3 kg of lead.

	KOSMOS 3.0	STAVIRO	FishOASIS	Opaleye	FishCam
**Reference**	This paper	[4,6]	[18]	[19]	[20]
**Sensor type**	12 Mp Raspberry PiCam HQ	SONY cameras (PJ740, CX900)	SONY α7s II camera	Logitech BRIO Webcam	8 MP Raspberry Pi Cam v2
**Image resolution**	1600 × 1200 pi	1920 × 1080 pi	4240 × 2380 pi	4k30	1600 × 1200 pi
**Field of View**	60° × 6 frames	60° × 6 frames	180°	65°, 78° or 90°	110°
**Storage**	USB 3.0 64 Go flash drive	128 Go Class 10 SD card	USB 3.0 256 Go flash drive	64 Go microSD card	200 Go microSD card
**Runtime**	~8 h	~8 h (camera), 3 days (motor)	224 h (16 h/day for 14 days)	External power source (wired)	212 h over 14 days
**Total weight**	~7 kg	~8 kg	Not specified	Not specified	Not specified
**Size**	300 mm length 101 mm diameter	Motor housing: 217 × 141 × 121 mm^3^ Camera housing: 300 mm length and 99 mm diameter	Not specified	165 mm length 127 mm diameter (enclosure)	300 mm length 101 mm diameter
**Cost**	€1360	€3250	€4260	€1350	€430
**Documentation for reproduction**	https://wikifactory.com/@konkarlab/kosmos30 (accessed on 14 October 2021)	Available from the lead author	https://github.com/cpagniel/FishOASIS/ (accessed on 14 October 2021)	https://github.com/suburbanmarine/opaleye (accessed on 14 October 2021)	https://github.com/xaviermouy/FishCam (accessed on 14 October 2021)
**Licence type**	CC-BY-SA 4.0	No licence	Not specified	Software: BSD-3-Clause license Hardware: CC-BY-SA 4.0	CC-BY-SA 4.0
**Existing applications**	Experiment in paper	>5000 valid videos (see Pelletier et al. 2021)	Experiment in paper	Not available	Experiment in paper
**Typical use**	Autonomous lander	Autonomous lander	Diver-placed fixed setup	To be integrated on a powered platform	Autonomous lander

## Data Availability

The data are available in the Appendix A and at https://wikifactory.com/@konkarlab/kosmos-30 (accessed on 14 October 2021).

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
