# Peer review of "KOSMOS: An Open Source Underwater Video Lander for Monitoring Coastal Fishes and Habitats"

_sensors, 2021, doi:10.3390/s21227724_

Round 1

Reviewer 1 Report

General Comments

This paper presents the development of an open-source underwater imaging system designed to be deployed for short durations of time to record video of fishes and the surrounding environment in shallow water. Of particular note is the incorporation of a mechanical system to rotate the field of view  in fixed increments, and a lander mechanical structure to facilitate straightforward deployment from the surface. The design leverages the extensive work on STAVIRO, but modernizes the design and dissemination of the methodology by adopting standard open hardware and software principles and leveraging imagers and compute systems from the Raspberry Pi foundation. The system presented here provides an essential data source for monitoring and the development and documentation provided can help accelerate progress in commoditizing this type of imaging system such that it is readily available for use in a wide range of applications

Open Source Documentation

I was hoping to see a link to a github page or similar with full BOMs and documentation for those interested in building a KOSMOS lander. Although perhaps not a requirement for publication, creating this resource can greatly enhance the dissemination and utilization of these kinds of systems. This is especially true for the software/firmware used for the system. 

System Specifications

I would encourage adding a table that lists the current specifications for the system such as storage, runtime, resolution, field of view, SWAP, and cost. Ideally, this table would also include a comparison other systems, some examples of which are: https://tos.org/oceanography/article/an-optical-imaging-system-for-capturing-images-in-low-light-aquatic-habitats-using-only-ambient-light , https://github.com/suburbanmarine/opaleye , https://www.sciencedirect.com/science/article/pii/S0967063719302821

Image Distortion and Aberrations (section 3.3)

It would be helpful to note here that lens distortion in the way it is described is a result of the non-ideal nature of the lens itself and when objects are imaged in air this distortion can be corrected and removed from the image. In contrast, imaging through a flat port underwater introduces additional distortion and breaks the single-view-point assumption typically applied in quantitative camera applications. For a full treatment of the topic this paper is an excellent resource: https://webee.technion.ac.il/~yoav/publications/flatrefracte.pdf. In addition to distortion, imaging through a flat port at a modestly wide angle also introduces lateral color separation away from the optical axis and field curvature. Both of these factors lower image quality near the corners of the field of view compared to the center.

Color Correction (section 5)

There is some discussion in section 5 about color correction as a function of depth. This is a major topic in and of itself and an ill-posed problem depending on the environmental conditions. Consider for example the difference in the downwelling radiance spectral distribution in clear water at 40 m vs high productivity coastal water at the same depth. Depth alone is insufficient to quantitatively correct color unless in pure water. Despite this, in the clear water case there have been some excellent papers recently that deal with the wavelength-dependent attenuation in dehazing such as: https://arxiv.org/abs/1811.01343. Such a method could readily be applied to these images.

Assessment

Readers are going to be interested in the contrast in the images in different lighting conditions and the image resolution. There is an obvious tradeoff in using a rasbi camera vs a $2k dslr in terms of dynamic range and light collection. Adding examples of in situ imaging at sunset and sunrise would be especially helpful to convince the reader that this is possible with the system, even in shallow, clear water. The test tank data could then be used to quantify spatial resolution vs. range in ideal underwater conditions.

Going further

Although single-camera underwater imaging systems can provided a great deal of information and context, they typically lack the ability to provide quantitative information about objects in the field of view. Given the low-cost and rapid advancement of Rasbi-based imaging systems (take a look at StereoPiV2 for example: https://www.stereopi.com/v2), I would include some discussion about how the KOSMOS design could be adapted to host stereo cameras on the rotating stage, or a set of stereo cameras to observe the full 360 panorama at the same time. Describing an architecture that could support these kinds of systems as pluggable building blocks would be especially useful to the community.

Author Response

Please see attached document that is easier to read

REPORT REVIEWER 1

General Comments

This paper presents the development of an open-source underwater imaging system designed to be deployed for short durations of time to record video of fishes and the surrounding environment in shallow water. Of particular note is the incorporation of a mechanical system to rotate the field of view  in fixed increments, and a lander mechanical structure to facilitate straightforward deployment from the surface. The design leverages the extensive work on STAVIRO, but modernizes the design and dissemination of the methodology by adopting standard open hardware and software principles and leveraging imagers and compute systems from the Raspberry Pi foundation. The system presented here provides an essential data source for monitoring and the development and documentation provided can help accelerate progress in commoditizing this type of imaging system such that it is readily available for use in a wide range of applications

Reply: Thank you for this positive assessment

 Open Source Documentation

I was hoping to see a link to a github page or similar with full BOMs and documentation for those interested in building a KOSMOS lander. Although perhaps not a requirement for publication, creating this resource can greatly enhance the dissemination and utilization of these kinds of systems. This is especially true for the software/firmware used for the system. 

Reply: Enhancing the dissemination and utilization of KOSMOS is definitely our intention. We set up a wiki page that is currently being translated to English. We have not yet created a github for the KOSMOS; thank you for this suggestion that we will soon implement (together with the English version of the documentation). 

System Specifications

I would encourage adding a table that lists the current specifications for the system such as storage, runtime, resolution, field of view, SWAP, and cost. Ideally, this table would also include a comparison other systems, some examples of which are:

https://tos.org/oceanography/article/an-optical-imaging-system-for-capturing-images-in-low-light-aquatic-habitats-using-only-ambient-light, https://github.com/suburbanmarine/opaleye, https://www.sciencedirect.com/science/article/pii/S0967063719302821

Reply: Thank you for these references. We created an additional table as suggested. We included another reference (Mouy et al. 2020). The table compares important features of the systems; we also added consideration of licence and use. We did not include Phillips et al. as the use of the camera differs a lot from the other systems. Two sentences were added at lines 578-581 to quote the table: “Several systems able to capture underwater video for observing biodiversity were proposed in recent years, that share similar low-cost and do-it-yourself objectives (Table 2). KOSMOS mostly differs from other propositions in that it is the successor of an existing system which used for a long time.”

Image Distortion and Aberrations (section 3.3)

It would be helpful to note here that lens distortion in the way it is described is a result of the non-ideal nature of the lens itself and when objects are imaged in air this distortion can be corrected and removed from the image. In contrast, imaging through a flat port underwater introduces additional distortion and breaks the single-view-point assumption typically applied in quantitative camera applications. For a full treatment of the topic this paper is an excellent resource: https://webee.technion.ac.il/~yoav/publications/flatrefracte.pdf. In addition to distortion, imaging through a flat port at a modestly wide angle also introduces lateral color separation away from the optical axis and field curvature. Both of these factors lower image quality near the corners of the field of view compared to the center.

Reply: We fully agree. We added a sentence to specify the origin of this distortion (lines 285-287):  “The optical distortion is both due to the non-ideal nature of the objective and to the use of a flat port. Imaging deformations associated to such optical system are well described [10] »

Thank you for the reference “Flat Refractive Geometry”; we will consider implementing such a model in forthcoming software developments.

Color Correction (section 5)

There is some discussion in section 5 about color correction as a function of depth. This is a major topic in and of itself and an ill-posed problem depending on the environmental conditions. Consider for example the difference in the downwelling radiance spectral distribution in clear water at 40 m vs high productivity coastal water at the same depth. Depth alone is insufficient to quantitatively correct color unless in pure water. Despite this, in the clear water case there have been some excellent papers recently that deal with the wavelength-dependent attenuation in dehazing such as: https://arxiv.org/abs/1811.01343 . Such a method could readily be applied to these images.

Reply: We agree. Actually, this paper is focused on the image acquisition system and its comparison with the parent system. Image post-processing, e.g. color correction or dehazing, is carried out after field work, once images are downloaded. Therefore, it is not part of this paper; we only wanted to mention that the images produced by KOSMOS may be readily corrected. This was clarified at lines 529-531 :

“the image format facilitates post-field image correction with respect to depth-dependent colour absorption (an interesting feature of the Paralenz camera) and turbidity [12–14], among other factors”

and the suggested reference was added at line 531.

Assessment

Readers are going to be interested in the contrast in the images in different lighting conditions and the image resolution. There is an obvious tradeoff in using a rasbi camera vs a $2k dslr in terms of dynamic range and light collection. Adding examples of in situ imaging at sunset and sunrise would be especially helpful to convince the reader that this is possible with the system, even in shallow, clear water. The test tank data could then be used to quantify spatial resolution vs. range in ideal underwater conditions.

Reply: We welcome this suggestion as lighting conditions and image resolution play a crucial role in image acquisition. The images showed in this paper were taken in controlled conditions in a pool and an aquarium for the purpose of showing the prototype performance and the close match to the parent system. The pool was equipped with two halogen projectors above tank and freshwater. More field work at sea is needed to appraise image restitution in varying conditions of light and visibility. This work is planned in the next months. We added a perspective sentence at lines 549-550:

“Future field work at sea will include testing the optical performance of the KOSMOS in varying conditions of visibility and light.”

An important motivation for selecting this camera was indeed to increase the system’s reproductibility over time. Commercial camera models regularly change and we had faced this issue in the past with the STAVIRO which uses commercial cameras. A challenge was thus to find a camera and lens producing quality images that are comparable between the KOSMOS and the STAVIRO. Additional constraints arose from the tube volume and the need to find a lens which size can adapt to the 2/3” camera sensor size.

Going further

Although single-camera underwater imaging systems can provided a great deal of information and context, they typically lack the ability to provide quantitative information about objects in the field of view. Given the low-cost and rapid advancement of Rasbi-based imaging systems (take a look at StereoPiV2 for example: https://www.stereopi.com/v2), I would include some discussion about how the KOSMOS design could be adapted to host stereo cameras on the rotating stage, or a set of stereo cameras to observe the full 360 panorama at the same time. Describing an architecture that could support these kinds of systems as pluggable building blocks would be especially useful to the community.

Reply: Indeed. The first objective of the KOSMOS is to provide a system that can capture images that are similar to the STAVIRO for which thousands of observations are available. In 2007, the choice of single camera was motivated by: 1) the need to conduct fast monitoring at sea, possibly with non-experts and from small boats (thus a light compact system), 2) the fact that analyzers trained with our protocol were able to assign enough information for our scientific needs (fish counts, fish ID and rough size classes per fish); and 3) the objective of keeping the overall cost reasonable (one camera, data storage and image analysis). In 2014, we successfully trialed a stereo version of the STAVIRO, but did not conduct our routine data collection this way.

For the KOSMOS, and given the technological solutions available now, we definitely intend to develop a stereo version, after this single-camera system. We added the following sentence at lines 551-556:

Last, the way the KOSMOS was conceived and made, and the rapid advancement of low-cost imaging technology (e.g. https://www.stereopi.com/v2) allow to envisage other developments such as the programmable version of the STAVIRO (MICADO)[15,16], and a stereo version that will provide size-based information and distance measurements. The MICADO is particularly relevant in places where visibility varies as it can record images at planned timings.”

Reviewer 2 Report

This manuscript provides an interesting overview of the KOSMOS Underwater video lander and builds on existing research by the authors.

Introduction

Introduction gives a very broad overview of coastal habitats as a whole - is there a target habitat or location etc which these systems have been implemented? If so, this should be made clearer in the introduction. Images within the paper suggest coral reef habitats?

When talking about benthic habitat monitoring, the visual techniques listed in lines 39 to 41 target epifaunal species and / or motile assemblages. Benthic grab sampling would be a key technique for benthic habitat monitoring but the macrofauna derived from this type of sampling would not be what would be sampled on video or diver observations. Instead of saying 'benthic habitats', it would be beneficial to list the type of faunal assemblages you are targeting.

Materials and Methods

Line 99 to 102 - With the references made to the observation of fishes at 5 - 10 m, is this system targeted towards certain habitats where visibility levels are high?

Discussion

Further detail on the practical application and conservation relevance of this system would be beneficial to this section. Can it for instance be used for monitoring assemblages around renewable developments or would the turbidity of there areas have an influence on its capabilities?

Is this a stand-alone monitoring tool or something which should be used in tandem with other techniques?

Author Response

Please see attached document that is easier to read

REPORT REVIEWER 2

This manuscript provides an interesting overview of the KOSMOS Underwater video lander and builds on existing research by the authors.

Thank you.

Introduction

Introduction gives a very broad overview of coastal habitats as a whole - is there a target habitat or location etc which these systems have been implemented? If so, this should be made clearer in the introduction. Images within the paper suggest coral reef habitats?

Reply: We added in the Introduction more information about the existing applications of the STAVIRO; the cited paper Pelletier et al. 2021 ([5] below) recaps all these existing data. The broad applicability of the protocol across habitats and ecosystems was clarified at lines 60-62:

“The STAVIRO protocol has proved efficient and reliable with over 5000 valid deployments in varied habitats of both temperate and tropical coral reef ecosystems since 2007 [5] »

When talking about benthic habitat monitoring, the visual techniques listed in lines 39 to 41 target epifaunal species and / or motile assemblages. Benthic grab sampling would be a key technique for benthic habitat monitoring but the macrofauna derived from this type of sampling would not be what would be sampled on video or diver observations. Instead of saying 'benthic habitats', it would be beneficial to list the type of faunal assemblages you are targeting.

Reply: We agree that the term “benthic habitat” can have several meanings. This video protocol targets the fish assemblage (and possibly other large motile animals turtles or snakes), and by benthic habitat, we mean biotic and abiotic covers, complexity and rugosity that can be described from underwater imagery.

We removed the term “Benthic habitat” at line 46 (mentioning only fishes). We defined this term at line 56-57 and added a reference using our benthic habitat data:

By benthic habitat, we mean here abiotic and biotic cover and related parameters [5]

Materials and Methods

Line 99 to 102 - With the references made to the observation of fishes at 5 - 10 m, is this system targeted towards certain habitats where visibility levels are high?

Reply: This system is intended for all habitats. Visibility may be poor in temperate ecosystems, e.g. in the Mediterranean and in the Atlantic Ocean, and even in tropical coral reef ecosystems, in particular when sampling soft bottom areas. In our protocol, a video is valid only if the estimated visibility is at least 5m. Images are systematically analyzed within a 5m radius around the camera, and if there are other animals seen beyond 5m, they are counted separately and not used in abundance indices.

Discussion

Further detail on the practical application and conservation relevance of this system would be beneficial to this section. Can it for instance be used for monitoring assemblages around renewable developments or would the turbidity of there areas have an influence on its capabilities?

Reply: Indeed, thank you for this comment. We have been using the parent system since 2007 first for assessing the ecological status of fish assemblages and benthic cover (references in Pelletier et al. 2021). But more recently, it has been suggested that this lander would be useful to monitor coastal developments such as MRE. In this case, a programmable system such as the MICADO would be needed to address varying visibilities that may occur depending on the areas surveyed.

We now quote the MICADO at line 553 and we added the sentence at lines 554-555:

“The MICADO is particularly relevant in places where visibility varies as it can record images at planned timings.”

Regarding the practical applications, we added the following sentence at lines 610-615:

“In the light of conservation challenges in coastal areas (marine renewable developments, fishing and other anthropogenic pressures), key biodiversity facets and fish resources must be monitored and assessed over large areas and with appropriate spatial replication. The ability to reproduce the KOSMOS at a reasonable cost is an opportunity to meet these requirements and collect numerous images that will be consistent with the existing STAVIRO images.”

Is this a stand-alone monitoring tool or something which should be used in tandem with other techniques?

Reply: So far, it has been used as a stand-alone tool, mostly because the primary aim was to be able to survey quickly and easily large areas and provide both spatial coverage and replicated data. There are two aspects in this question: i) in recent years, the developments of compact sensors enables to envisage having additional sensors on the lander – e.g. add acoustic sensors that would ideally complement images, or sensors for physical parameters; ii) it is also possible to devise a mixed sampling design that combines the deployments with other monitoring tools such as Underwater Visual Censuses (we conducted an intercalibration experiment, see Mallet et al. 2014), Baited Remote Underwater Video, catch-based samples or even e-DNA.  
